# Induced abortion incidence and safety in Côte d'Ivoire

**Suzanne O. Bell**[1]*, **Grace Sheehy**[1], **Andoh Kouakou Hyacinthe**[2], **Georges Guiella**[3], **Caroline Moreau**[1,4]

1 Department of Population, Family and Reproductive Health, The Johns Hopkins Bloomberg School of Public Health, Baltimore, Maryland, United States of America, 2 Programme National de Santé de la Mère et de l'Enfant (PNSME), Abidjan, Côte d'Ivoire, 3 Institut Supérieur des Sciences de la Population (ISSP), Université of Ouagadougou, Ouagadougou, Burkina Faso, 4 Soins Primaires et Prévention, CESP Centre for Research in Epidemiology and Population Health, U1018, Inserm, Villejuif, France

* suzannebell@jhu.edu

## Abstract

### Background

In Côte d'Ivoire, induced abortion is legally restricted unless a pregnancy threatens a woman's life. Yet the limited available evidence suggests abortion is common and that unsafe abortion is contributing to the country's high maternal mortality. Our study aimed to estimate the one-year incidence of induced abortion in Côte d'Ivoire using both direct and indirect methodologies, determine the safety of reported abortions, and identify the women most likely to experience a recent induced abortion or an unsafe abortion.

### Methods

In 2018, we conducted a nationally representative, population-based survey of women age 15 to 49 in Côte d'Ivoire. Women reported their own abortion experiences and those of their closest female confidante. We estimated the one-year incidence of induced abortion, and the safety of the abortions women experienced. Using bivariate and multivariate regression, we separately assessed sociodemographic characteristics associated with having had a recent abortion or an unsafe abortion.

### Results

Overall, 2,738 women participated in the survey, approximately two-thirds of whom reported on the abortion experiences of their closest female friend. Based on respondent data, the one-year incidence of induced abortion was 27.9 (95% CI 18.6–37.1) per 1,000 women of reproductive age, while the confidante incidence was higher at 40.7 (95% CI 33.3–48.1) per 1,000. Among respondents, 62.4% of abortions were most unsafe, while 78.5% of confidante abortions were most unsafe. Adolescents, less educated women, and the poorest women had the most unsafe abortions.

**Data Availability Statement:** Data for this study are publicly available and can be requested online at pmadata.org. The authors used the Cote d'Ivoire

Round 2 household/female dataset for this analysis.

**Funding:** All funding for this study provided by an Anonymous Donor (grant number 127941). The funders had no role in study design, data collection and analysis, decision to publish, or preparation of the manuscript, and the authors are not aware of any donor competing interests.

**Competing interests:** The authors have declared that no competing interests exist.

## Conclusion

This study provides the first national estimates of induced abortion incidence and safety in Côte d'Ivoire, using a population-based approach to explore social determinants of abortion and unsafe abortion. Consistent with other research, our results suggest that legal restrictions on abortion in Côte d'Ivoire are not preventing women from having abortions, but rather pushing women to use unsafe, potentially dangerous abortion methods. Efforts to reduce the harms of unsafe abortion are urgently needed.

## Introduction

The West African country of Côte d'Ivoire has a relatively young population and a fast-growing economy; however, a decade of conflict and instability that destroyed 42% of health facilities weakened the health system and has contributed to poor population health indicators [1]. Use of effective methods of contraception among women of reproductive age is low, with a modern contraceptive prevalence of 20.9%, and more than one-third (34.9%) of women reporting their last pregnancy was unintended [2]. The total fertility rate remains high at five children per woman and has declined only minimally in recent years [3]. Further, the country has one of the highest maternal mortality rates in the region, with estimates ranging from 502 to 944 deaths per 100,000 live births [4, 5]. These deaths are in large part a result of limited emergency obstetric care, but unsafe abortion is a significant contributor.

In Côte d'Ivoire, induced abortion is legally restricted unless a pregnancy threatens a woman's life. The country's Penal Code states that two medical providers must examine a woman and agree that an abortion is necessary to save her life before a pregnancy can be legally terminated [6]. Despite a lack of evidence of enforcement, anyone who provides or assists in providing an abortion—as well as the woman who obtains an abortion herself—can be punished under the law with a prison sentence and fine [7]. Although there are no national estimates of the abortion rate in Côte d'Ivoire, available evidence suggests that abortion, particularly unsafe abortion, is common.

West Africa has some of the highest rates of unsafe abortion in the world, with estimates indicating as many as 85% of abortions in the region are unsafe, which the World Health Organization (WHO) defines as being performed by an individual lacking the necessary training or in an environment not conforming to minimal medical standards [8]. These unsafe abortions are subsequently responsible for 10 to 18% of maternal deaths [5, 9]. Despite these risks, existing research suggests that abortion is a common means of fertility control in Côte d'Ivoire. In one national survey, 42.5% of women of reproductive age with a history of pregnancy reported having had a prior abortion [10]. The same study found that 50.1% of abortions reported took place in the home, using methods such as plants, while 47.9% took place in a health facility. Nearly half (49.4%) of reported abortions were performed by traditional practitioners or women themselves. Most women who had an abortion were under the age of 25 and unmarried. Based on these limited available data, the majority of abortions in Côte d'Ivoire are likely to be unsafe.

While unsafe abortion is a significant contributor to maternal morbidity and mortality, the measurement of abortion is challenging due to underreporting in facility statistics, omission of abortions that take place outside the healthcare system, and underreporting in self-report questions in population-based surveys. Research on women's abortion experiences outside the formal healthcare system, particularly self-managed medication abortion, is scarce and

typically not representative. Capturing these experiences is essential to understanding the scope and determinants of abortion and of unsafe abortion in Côte d'Ivoire.

The first aim of this study was to estimate the one-year incidence of induced abortion in Côte d'Ivoire overall and by women's background characteristics, using both direct and indirect methodologies. Our second aim was to determine the safety of reported abortions and identify the women who were most likely to experience the most unsafe abortions.

## Methods

### Sampling

Data for this study come from the population-based survey of reproductive age (15–49) women in Côte d'Ivoire conducted by Performance Monitoring and Accountability 2020 (PMA2020)/Côte d'Ivoire. PMA2020 is a multi-disciplinary team of researchers that conducts rapid-turnaround survey-based research in 11 countries using smart phones. The Institut National de la Statistique de la Côte d'Ivoire (INS-Côte d'Ivoire) and the Coordination du Programme National de Santé de la Mère et de l'Enfant (DC-PNSME) within the Ministry of Health implemented the PMA2020/Côte d'Ivoire abortion project with guidance from the Institut Supérieur des Sciences de la Population in Ouagadougou, Burkina Faso and overall direction and support provided by the Bill & Melinda Gates Institute for Population and Reproductive Health at the Johns Hopkins Bloomberg School of Public Health.

These data were collected as part of the second round of PMA2020/Côte d'Ivoire, which occurred from July through August 2018. The sampling strategy relied on an urban-rural stratified cluster design with probability proportional to size selection of 73 enumeration areas (EAs), each of which represented a cluster of approximately 200 households. The National Statistics Institute (INS) selected the EAs from a sampling frame provided by the 2014 General Census of Population and Housing. In each EA, female resident interviewers mapped and listed all households and supervisors randomly selected 35 households from each EA sampling frame created. All women age 15 to 49 identified in selected households were eligible to participate in the face-to-face surveys, which interviewers conducted in French or local languages using smartphones, entering data via an Open Data Kit (ODK) application on the phone; the English and the French translation of the questionnaire are provided in the supplementary materials (S1 Doc and S2 Doc). In order for the data to be nationally representative we constructed survey weights, which we calculated using the inverse of the probability of selection, accounting for the probability of EA selection, probability of household selection, and household and female response rates. The final sample included 2,738 *de facto* women (female response rate 98.1%). The Johns Hopkins Bloomberg School of Public Health institutional review board (IRB) and the Comité d'Éthique de la Recherche of Côte d'Ivoire provided ethical approval for this study. Women provided verbal informed consent prior to participation, with minors treated as adults in accordance with local IRB approval. Interviewers indicated receipt of verbal consent by checking a box in the smartphone survey to confirm and entering their name as a witness to the consent process.

### Measures

The household survey produced data on household wealth while the linked female survey covered socio-demographic characteristics, current and past pregnancies, contraceptive knowledge and use, and related reproductive health topics. All women who consented to participate in the core PMA2020 female questionnaire also answered questions in the abortion module, which explored abortion frequency, methods, and sources in Côte d'Ivoire. The female

abortion module included additional questions related to the respondent and her closest confidante's experiences with abortion.

In the abortion module, the interviewer first asked the respondent to indicate the number of close female confidantes she had, defining a close female confidante as a woman age 15 to 49 currently living in Côte d'Ivoire who shares personal information with the respondent and with whom the respondent shares personal information. The respondent provided the age and education of their closest female confidante before the interviewer asked about the confidante's experience with abortion. No prior questions in the survey had mentioned abortion so as not to bias respondents' confidante selection. This indirect confidante approach builds off prior social network-based measurement of abortion [11–13]. We write about this method in more detail elsewhere [14].

Terminology and question phrasing are particularly important when addressing a sensitive topic like induced abortion. In order to avoid inclusion of miscarriage experiences, interviewers read the following preamble at the outset of the abortion module to indicate that subsequent questions were in the context of an unintended pregnancy: "Sometimes women are worried they are pregnant or get pregnant when they do not want to be and they do something to remove the pregnancy." To assess respondent interpretation, the pilot questionnaires included quantitative face validity questions where the interviewer asked the respondent to describe how she interpreted or understood the phrases "pregnancy removal" and "period regulation at a time when you were worried you were pregnant". In total, interviewers conducted 31 pilot surveys in Cote d'Ivoire. Interviewers indicated that 100% of pilot survey respondents interpreted the "pregnancy removal" and "period regulation" phrases correctly.

After collecting details on the respondent's confidantes, the interviewer asked separate questions of the respondent regarding the confidante's experience with pregnancy removal at a time the confidante was pregnant or worried she was pregnant, or period regulation at a time the confidante was worried she was pregnant. The interviewer then collected additional details about reported pregnancy removals and period regulations, including year, whether the woman did multiple things to terminate the pregnancy, method(s), and source(s). We were unable to collect information on repeated abortions thus these details correspond to the most recent pregnancy removal or period regulation. For confidantes who did multiple things in the process of terminating, subsequent questions inquired about the first method and source and the last method and source. The interviewer then asked these questions with regard to the respondent's own experience. Abortion methods included surgery, medication abortion (MA) drugs, other pills (antibiotics, antimalarial medication) or unspecified pills, and traditional or other methods (industrial products like bleach, herbal remedies, inserting materials into the vagina). Sources included public facilities, private facilities (including non-governmental organizations and private doctors), pharmacies or chemist shops, and traditional or other sources (including shops, markets, friends or relatives, or home). Interviewers did not read method and source options aloud; women volunteered their answer and interviewers probed when necessary to determine the appropriate response option to select. In the pilot, women were unable to provide detail regarding the specific surgery type (e.g. dilation and curettage, manual vacuum aspiration), thus interviewers only selected the one "surgery" option if a woman described having a procedure.

We operationalized abortion safety using the method and source information, corresponding to two dimensions of safety: 1) whether the process involved any non-recommended methods (i.e. other than surgery or medication abortion drugs) that put the woman at potentially high risk of abortion-related morbidity or mortality; and 2) whether the source(s) used were clinical (public or private facilities) or non-clinical (any other source). For respondents or confidantes who did more than one thing to terminate the pregnancy, we categorized their

abortion as involving a non-recommended method if either the first or last method was something other than surgery or MA drugs. Similarly, we categorized an abortion as involving a non-clinical source if at any point the woman accessed care outside of a public or private facility. To create a single measure of safety, we combined these two dimensions to create the following four categories of abortion safety: 1) recommended method(s) involving only clinical source(s); 2) recommended method(s) involving non-clinical source(s); 3) non-recommend method(s) involving only clinical source(s); and 4) non-recommended method(s) involving non-clinical source(s). We deemed abortions in group four as the most unsafe [15].

## Analysis

We conducted univariate analyses of respondent and confidante characteristics. Due to questionnaire length constraints, we only collected information on confidantes' age and education. We made a number of adjustments to the confidante data to improve the validity of the abortion incidence and safety estimates. The confidante estimates included all confidante pregnancy removals and period regulations that respondents reported with certainty (response option "Yes, I am certain") or less certainty (response option "Yes, I think so") when they could still provide the specific method(s) the confidante used. Inclusion of the less certain abortions helped to counteract respondent's incomplete knowledge of their confidante's abortions (i.e. transmission bias). We also adjusted the confidante estimates for potential selection bias resulting from the fact that some respondents reported zero confidantes. For this adjustment, we ran a Poisson model to predict the likelihood of these "missing" confidantes having had an abortion in the prior year. This model regressed the socioeconomic characteristics of the confidantes and the respondents with no confidantes on the available confidante abortion incidence data. We then used the model to predict the likelihood of the "missing" confidantes having had a recent abortion based on the corresponding respondents' characteristics. We used this information to create a new variable that combined respondent reported confidante abortion data for those with confidantes, and the predicted probability of abortion in the prior year for the confidantes who were not in the sample because they had no close friends who we could have captured in the respondent sample. Research on mortality rate estimation using survey data has employed a similar modeling approach [16]. To ensure these confidante data had characteristics that reflected the population of reproductive aged women in Cote d'Ivoire, we constructed post-stratification weights using the weighted respondent data distributions as the reference.

We separately calculated the one-year pregnancy removal and period regulation incidence rates. We were unable to collect data on month of the event due to questionnaire length constraints, thus we included events from 2017 through the date of interview in 2018. To convert this to an annualized one-year incidence rate, we divided the number of events in 2017 and 2018 by the number of woman-years between January 1, 2017 through the date of the interview in 2018; each respondent contributed on average 1.55 woman-years. We then multiplied the value by 1,000 to generate the approximate one-year incidence rate per 1,000 women age 15 to 49. We scaled the standard errors in the same manner. We also calculated the combined pregnancy removal and period regulation one-year incidence rate (which we refer to as "likely-abortion"). To calculate the final one-year incidence rates of induced abortion, we averaged the pregnancy removal rate and the likely-abortion rate. We averaged the two point estimates because we believe the pregnancy removal data fails to capture some abortions (that women may not view as abortions or are not willing to admit are abortions) while the period regulation data likely include some experiences that we would not consider to be abortions. We generated all these estimates separately for respondents and confidantes.

We calculated one-year pregnancy removal incidence rates, likely-abortion incidence rates, and the associated averages overall and by age and education for respondents and confidantes; we also calculated these rates by residence and wealth for respondents, for whom we had data on these characteristics. The final incidence analyses involved bivariate and multivariable logistic regression to determine which characteristics were independently associated with experiencing a recent likely-abortion in the approximately one year prior to the survey (2017 through beginning of 2018). We used the unadjusted confidante likely-abortion dichotomous incidence data in order to conduct the logistic regression analyses as the model assumes a Bernoulli distributed outcome variable whereas the Poisson predicted confidante incidence variable is continuous.

The abortion safety analyses first assessed the overall distribution of safety for respondents and confidantes along the two dimensions previously described. We separately estimated the proportion of respondents and confidantes who experienced the most unsafe abortions (i.e. those involving non-recommended methods from non-clinical sources) overall and by age and education for both respondents and confidantes, and residence and wealth for respondents only. We then conducted bivariate and multivariable logistic regressions to examine which characteristics were independently associated with increased odds of experiencing a most unsafe abortion. The safety analyses do not include any Poisson imputed safety data, unlike the incidence calculations. For the final analysis we estimated the one-year incidence rate of the most unsafe abortions and the corresponding annual number of most unsafe abortions in Côte d'Ivoire.

We conducted all analyses in Stata version 15.1 [17]. We weighted results using the Taylor linearization method and calculated standard errors using a robust variance estimator to account for the complex sampling design and clustering, respectively.

## Results

### Sample characteristics

The final sample included 2,738 respondents, 64.1% of whom reported having at least one close female confidante, resulting in a confidante sample of 1,756 women (Table 1). The average age of respondents was 28.5, and the majority had little formal schooling, with 45.2% having never attended school and 25.9% having attended primary school. Approximately two-thirds (64.8%) of respondents were currently married or cohabitating. The majority of women were religious (88.9%), primarily identifying as Muslim (39.5%) or Catholic (20.3%). Respondents represented a range of ethnic groups, with the largest proportion identifying as Akan (34.6%), 20.8% as a non-Ivoirian ethnicity, and 20.8% as Mande. One-quarter of participants had no children, while 32.2% had 1 to 2 children, 21.5% had 3 to 4 children, and 20.6% had five or more children. The majority of participants resided in urban parts of the country (61.5%). Confidantes were not significantly different from respondents in terms of age and education level.

### Abortion incidence

The overall one-year likely-abortion incidence (pregnancy removal and period regulation combined) in Côte d'Ivoire was 36.9 (95% confidence interval (CI) 25.4–48.5) per 1,000 women of reproductive age when using respondent data. The adjusted confidante likely-abortion incidence was higher at 50.0 (95% CI 41.9–58.1) per 1,000 women. Excluding the reported period regulations, the pregnancy removal incidence for respondents and confidantes were 18.8 (95% CI 11.8–25.8) and 31.5 (95% CI 24.8–38.1) for respondents and confidantes, respectively. Averaging these estimates, we calculated a final one-year induced abortion incidence of

**Table 1. Characteristics of female respondents age 15 to 49 and their closest female confidantes age 15 to 49 in Côte d'Ivoire[1].**

| | | Respondent | | Unadjusted Confidante | | Adjusted Confidante[2] | |
|---|---|---|---|---|---|---|---|
| | | N | % | N | % | N | % |
| Mean age | | 2,738 | 28.5 | 1,756 | 29.0 | 2,738 | 28.8 |
| Age | | | | | | | |
| | 15–19 | 542 | 20.1 | 305 | 17.9 | 484 | 19.0 |
| | 20–24 | 500 | 18.1 | 307 | 17.9 | 481 | 17.8 |
| | 25–29 | 495 | 17.9 | 298 | 16.0 | 470 | 17.2 |
| | 30–34 | 436 | 16.3 | 306 | 18.3 | 462 | 17.2 |
| | 35–39 | 351 | 12.8 | 255 | 13.6 | 370 | 12.7 |
| | 40–44 | 262 | 9.4 | 166 | 9.4 | 275 | 9.7 |
| | 45–49 | 152 | 5.5 | 119 | 6.9 | 196 | 6.4 |
| Education | | | | | | | |
| | Never | 1,254 | 45.2 | 773 | 42.8 | 1,267 | 45.3 |
| | Primary | 714 | 25.9 | 366 | 20.7 | 621 | 24.7 |
| | Secondary | 615 | 23.0 | 484 | 28.2 | 672 | 23.9 |
| | Higher | 152 | 6.0 | 134 | 8.3 | 176 | 6.5 |
| Marital status | | | | | | | |
| | Currently married/cohabiting | 1,767 | 64.8 | – | – | – | – |
| | Divorced or separated/widowed | 126 | 4.4 | – | – | – | – |
| | Never married | 844 | 30.8 | – | – | – | – |
| Religion of household | | | | | | | |
| | Muslim | 1,148 | 39.5 | – | – | – | – |
| | Catholic | 544 | 20.3 | – | – | – | – |
| | Evangelical | 406 | 15.4 | – | – | – | – |
| | Other | 382 | 13.7 | – | – | – | – |
| | No religion | 258 | 11.1 | – | – | – | – |
| Ethnicity of household | | | | | | | |
| | Akan | 889 | 34.6 | – | – | – | – |
| | Mande (nord and sud) | 575 | 20.8 | – | – | – | – |
| | Gur | 404 | 14.4 | – | – | – | – |
| | Other Ivoirian | 274 | 9.3 | – | – | – | – |
| | Other non-Ivoirian | 594 | 21.0 | – | – | – | – |
| Parity | | | | | | | |
| | 0 | 704 | 25.8 | – | – | – | – |
| | 1–2 | 867 | 32.2 | – | – | – | – |
| | 3–4 | 590 | 21.5 | – | – | – | – |
| | 5+ | 572 | 20.6 | – | – | – | – |
| Residence | | | | | | | |
| | Rural | 1,062 | 38.5 | – | – | – | – |
| | Urban | 1,676 | 61.5 | – | – | – | – |
| Mean number of confidantes | | 2,720 | 0.8 | – | – | – | |
| Total | | 2,738 | 100.0 | 1,761 | 100.0 | 2,738 | 100.0 |

27.9 (95% CI 18.6–37.1) per 1,000 women of reproductive age based on self-reports and 40.7 (95% CI 33.3–48.1) per 1,000 based on confidante data. The subsequent results are based on the average of pregnancy removal and pregnancy removal/period regulation incidences, which we simply refer to as abortion.

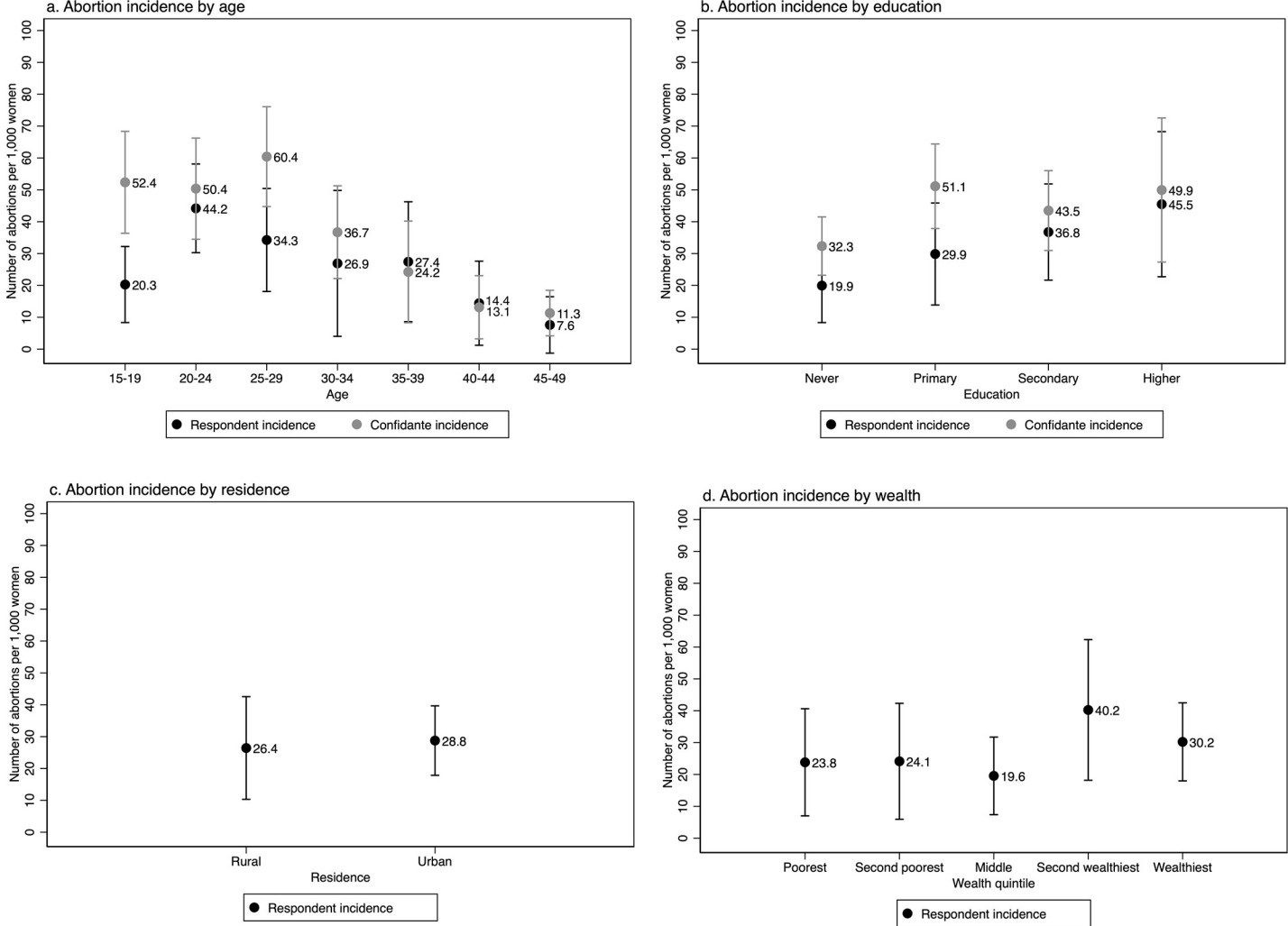

**Fig 1. One-year incidence of abortion (average of pregnancy removal and period regulation rates) per 1,000 women ag 15 to 49 among female respondents and their closest female confidantes in Côte d'Ivoire by background characteristics.**

Respondents 20 to 24 years old had the highest one-year abortion incidence at 44.2 abortions per 1,000, while confidantes 25 to 29 years old had the highest rates at 60.4 per 1,000 (Fig 1). Higher levels of education were associated with higher one-year abortion incidence among respondents: women with higher than a secondary education had an abortion incidence of 45.5, those with secondary schooling had an incidence of 36.8, and those with primary education had an incidence of 29.9. Trends differed somewhat for confidantes, with the highest incidence observed both among those who attended higher education (49.9), and those who had attended primary school (51.1). For both groups, the lowest abortion incidence was among women with no formal education, at 19.9 among respondents and 32.3 among confidantes. For respondents, the likelihood of abortion did not differ by residence, with 26.4 abortions per 1,000 in rural areas compared to 28.8 in urban areas. Additionally, wealthier respondents were more likely to have had a recent abortion. Examining the ratio of confidante to respondent abortion incidences, women age 15 to 19 were the least likely to report their own abortion experiences while women with primary education or less were less likely to report than women with higher levels of education.

**Table 2. Bivariate and multivariate regressions of characteristics associated with experiencing a recent likely-abortion among Côte d'Ivoire respondents and confidantes age 15 to 49[1].**

| | | Respondent (n = 2,733) | | | | | | Confidante (n = 1,760) | | | | | |
|---|---|---|---|---|---|---|---|---|---|---|---|---|---|
| | | OR | 95% CI | | aOR | 95% CI | | OR | 95% CI | | aOR | 95% CI | |
| Age | | | | | | | | | | | | | |
| | 15–19 | 1.00 | – | – | 1.00 | – | – | 1.00 | – | – | 1.00 | – | – |
| | 20–24 | **2.03** | **1.22** | **3.38** | **2.14** | **1.29** | **3.57** | 0.84 | 0.37 | 1.92 | 0.80 | 0.34 | 1.91 |
| | 25–29 | 1.61 | 0.97 | 2.69 | **1.77** | **1.03** | **3.03** | 1.19 | 0.68 | 2.10 | 1.08 | 0.56 | 2.07 |
| | 30–34 | 1.09 | 0.44 | 2.72 | 1.24 | 0.52 | 2.97 | 0.58 | 0.26 | 1.30 | 0.53 | 0.22 | 1.25 |
| | 35–39 | 1.35 | 0.57 | 3.19 | 1.52 | 0.64 | 3.62 | 0.34 | 0.09 | 1.32 | 0.31 | 0.07 | 1.27 |
| | 40–44 | 0.89 | 0.41 | 1.95 | 1.04 | 0.48 | 2.27 | 0.29 | 0.08 | 1.09 | 0.26 | 0.06 | 1.09 |
| | 45–49 | 0.52 | 0.15 | 1.84 | 0.59 | 0.17 | 2.01 | **0.06** | **0.01** | **0.45** | **0.05** | **0.01** | **0.45** |
| Education | | | | | | | | | | | | | |
| | Never | 1.00 | – | – | 1.00 | – | – | 1.00 | – | – | 1.00 | – | – |
| | Primary | 1.45 | 0.79 | 2.66 | 1.35 | 0.76 | 2.40 | 1.56 | 0.80 | 3.08 | 1.57 | 0.78 | 3.16 |
| | Secondary | 1.82 | 0.99 | 3.33 | **1.86** | **1.11** | **3.14** | 1.20 | 0.63 | 2.27 | 0.98 | 0.48 | 1.99 |
| | Higher | **2.42** | **1.15** | **5.11** | **2.38** | **1.19** | **4.75** | 1.57 | 0.69 | 3.53 | 1.42 | 0.62 | 3.24 |
| Residence | | | | | | | | | | | | | |
| | Rural | 1.00 | – | – | 1.00 | – | – | – | – | – | – | – | – |
| | Urban | 0.96 | 0.46 | 2.02 | 0.65 | 0.36 | 1.21 | – | – | – | – | – | – |
| Wealth quintile | | | | | | | | | | | | | |
| | Poorest | 1.00 | – | – | 1.00 | – | – | – | – | – | – | – | – |
| | Second poorest | 1.00 | 0.57 | 1.74 | 1.03 | 0.58 | 1.80 | – | – | – | – | – | – |
| | Middle | 0.91 | 0.34 | 2.48 | 1.00 | 0.40 | 2.50 | – | – | – | – | – | – |
| | Second wealthiest | 1.48 | 0.56 | 3.91 | 1.70 | 0.76 | 3.80 | – | – | – | – | – | – |
| | Wealthiest | 1.19 | 0.49 | 2.88 | 1.18 | 0.52 | 2.64 | – | – | – | – | – | – |

[1]Bolding indicates statistical significance at the p<0.05 level

In our logistic regressions, we found that being age 20 to 24 (compared to those age 15 to 19) was significantly associated with greater abortion incidence for respondents, while older confidantes (45 to 49) were significantly less likely to have had a recent abortion (Table 2). Greater education was positively associated with recent abortion for respondents and remained so in the multivariable regression while this factor did not rise to the level of significance for confidantes (Table 2). Place of residence (i.e. rural or urban) and wealth were not significantly associated with abortion incidence for respondents.

## Abortion safety

The majority of likely-abortions (pregnancy removal and period regulation combined) reported by women in the study were unsafe. Among respondents, 62.4% had likely-abortions that would be categorized as most unsafe (involving non-recommended methods and non-clinical providers), while 78.5% of confidantes had most unsafe likely-abortions (Table 3). Approximately one-third (32.7%) of respondents had the most safe likely-abortions, involving recommended methods and a clinical provider, while only 18.2% of confidante reported abortions were classified as most safe. Very few respondents or confidantes used recommended methods with a non-clinical provider (3.0% and 2.4%) or non-recommended methods with a clinical provider (1.9% and 0.9%). Among likely-abortions reported in the last five years, a larger percentage of respondent and confidante likely-abortions were considered most unsafe; 71.5% and 79.9%, respectively (estimates not shown).

**Table 3. Safety of most recent reported likely-abortion among female respondents age 15 to 49 and their closest female confidantes age 15 to 49 in Côte d'Ivoire.**

| | | Respondent | | Confidante | |
|---|---|---|---|---|---|
| | | Estimate | N | Estimate | N |
| Recommended method, clinical provider | | 32.7 | 198 | 18.2 | 75 |
| Recommended method, non-clinical provider | | 3.0 | 21 | 2.4 | 11 |
| Non-recommended method, clinical provider | | 1.9 | 18 | 0.9 | 6 |
| Non-recommended method, non-clinical provider | | 62.4 | 408 | 78.5 | 322 |
| Total | | 100.0 | 645 | 100.0 | 414 |

Abortion safety varied by respondent and confidante sociodemographic characteristics (Fig 2). Among respondents, the youngest women (age 15 to 19) had the most unsafe likely-abortions (78.0%), while among confidantes the older women had the most unsafe likely-abortions (91.3% among women 40 to 44). Among both respondents and confidantes, those with no formal education were most likely to have had the most unsafe likely-abortion (72.9% and 88.9%), compared to those with higher education who had the lowest levels of most unsafe (46.2% and 56.7%). Women in rural settings were more likely than those in urban settings to have the most unsafe likely-abortions, with 74.9% of respondents in rural parts of the country having the most unsafe likely-abortions. Based on respondent household wealth data, the poorest women were also most likely to have the least safe likely-abortions (80.1%); the proportion of likely-abortions categorized as most unsafe decreased steadily, with the wealthiest women least likely to have a most unsafe likely-abortion (44.4%).

In the multivariable analysis conducted among respondents reporting a likely-abortion, wealth remained significantly associated with unsafe abortion, while age, education and residence were no longer statistically significant (Table 4). In the confidante regressions, age and education were significantly associated with unsafe likely-abortion in both the bivariate and multivariate analyses, with increasing educated associated with reduced likelihood of having a most unsafe likely-abortion and older age and adolescence associated with increased likely of having a most unsafe likely-abortion (Table 4).

## Discussion

This study provides the first national estimates of induced abortion incidence and safety in Côte d'Ivoire. In line with the limited available evidence on abortion pathways in the country, our findings indicate that abortion in Côte d'Ivoire is common and predominantly takes place using non-recommended methods and performed by untrained providers. Our respondent findings indicate there are 27.9 abortions per 1,000 women of reproductive age while the confidante data suggest a higher annual rate of 40.7. Given concerns about underestimation in self-reported data, we believe the confidante incidence is closer to the true rate. Our final 2017 national incidence of 40.7 is just outside the uncertainty range of regional abortion incidence estimates for West Africa from 2010–2014 (31, 90% uncertainty interval 28–39) [18]. Using our incidence estimate and data on the Côte d'Ivoire population, we estimate there were more than 230,000 induced abortions in 2017, the majority of which were unsafe (62.4% for respondents and 78.5% for confidantes). Since more complicated abortions are likely to be more visible to one's social network, we view the confidante safety estimate as an overestimate. Thus, the respondent finding that 62.4% of abortions were most unsafe is more accurate. This

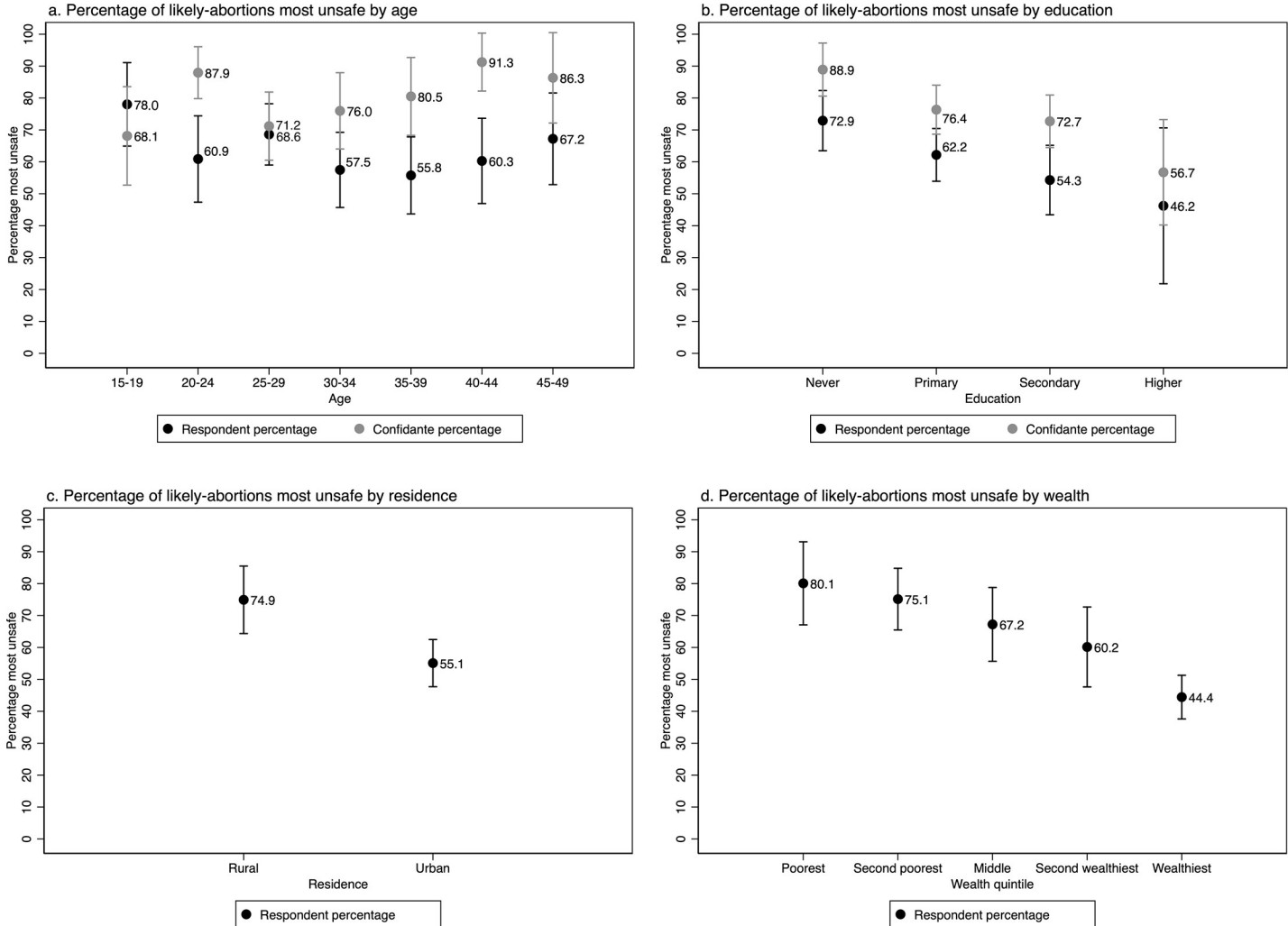

**Fig 2. Percentage of most recent likely-abortion among female respondents and their closest female confidantes in Côte d'Ivoire that were the most unsafe by background characteristics.**

estimate is between the WHO regional least safe (52.1%) and unsafe (84.7%) abortion estimates for West Africa, although our measurement of abortion safety differed [8]. However, the Cote d'Ivoire safety distribution was nearly identical to recent findings from Nigeria, where 63.4% of abortions were most unsafe [15].

Young women (under 30) and women with more education had the highest rates of abortion, while adolescents (age 15–19), less educated women, and the poorest women had the most unsafe abortions. Similar to our findings, Vroh and colleagues (2012) found higher abortion prevalence among women under the age of 25 in their 2007 cross-sectional study in Côte d'Ivoire. However, the authors also found higher occurrence of abortion among women with lower levels of literacy and women residing in urban parts of the country, which are in contrast to our findings. This study only had direct reporting of respondent's prior abortion experiences, thus their estimates may suffer from differential underreporting more so than our confidante data. In another study of women who had been admitted to gynecological departments, investigators similarly found that the majority of abortions were performed using unsafe abortion methods outside of clinical settings [19]. As a whole, these findings suggest that access to

**Table 4. Bivariate and multivariate regression of characteristics associated with experiencing a most unsafe likely-abortion among Côte d'Ivoire respondents and confidantes age 15 to 49[1].**

| | | Respondent (n = 645) | | | | | | Confidante (n = 414) | | | | | |
|---|---|---|---|---|---|---|---|---|---|---|---|---|---|
| | | OR | 95% CI | | aOR | 95% CI | | OR | 95% CI | | aOR | 95% CI | |
| Age | | | | | | | | | | | | | |
| | 15–19 | 1.00 | – | – | 1.00 | – | – | 1.00 | – | – | 1.00 | – | – |
| | 20–24 | 0.44 | 0.19 | 1.03 | 0.51 | 0.22 | 1.20 | **3.41** | **1.19** | **9.81** | **3.81** | **1.26** | **11.52** |
| | 25–29 | 0.62 | 0.29 | 1.32 | 0.67 | 0.30 | 1.47 | 1.16 | 0.49 | 2.71 | 1.20 | 0.47 | 3.07 |
| | 30–34 | **0.38** | **0.17** | **0.88** | 0.42 | 0.17 | 1.00 | 1.48 | 0.55 | 3.99 | 1.33 | 0.49 | 3.65 |
| | 35–39 | **0.36** | **0.15** | **0.84** | 0.42 | 0.17 | 1.04 | 1.93 | 0.67 | 5.60 | 1.94 | 0.69 | 5.46 |
| | 40–44 | 0.43 | 0.16 | 1.11 | 0.50 | 0.19 | 1.32 | **4.88** | **1.15** | **20.80** | **5.29** | **1.25** | **22.29** |
| | 45–49 | 0.58 | 0.21 | 1.61 | 0.48 | 0.19 | 1.22 | 2.95 | 0.81 | 10.80 | 1.78 | 0.48 | 6.64 |
| Education | | | | | | | | | | | | | |
| | Never | 1.00 | – | – | 1.00 | – | – | 1.00 | – | – | 1.00 | – | – |
| | Primary | **0.61** | **0.37** | **1.00** | 0.66 | 0.41 | 1.07 | 0.40 | 0.15 | 1.05 | **0.39** | **0.16** | **0.99** |
| | Secondary | **0.44** | **0.26** | **0.76** | 0.62 | 0.37 | 1.06 | **0.33** | **0.13** | **0.84** | **0.33** | **0.13** | **0.80** |
| | Higher | **0.32** | **0.11** | **0.96** | 0.60 | 0.21 | 1.76 | **0.16** | **0.06** | **0.43** | **0.14** | **0.06** | **0.35** |
| Residence | | | | | | | | | | | | | |
| | Rural | 1.00 | – | – | 1.00 | – | – | – | – | – | – | – | – |
| | Urban | **0.41** | **0.21** | **0.79** | 0.91 | 0.42 | 2.01 | – | – | – | – | – | – |
| Wealth quintile | | | | | | | | | | | | | |
| | Poorest | 1.00 | – | – | 1.00 | – | – | – | – | – | – | – | – |
| | Second poorest | 0.75 | 0.34 | 1.67 | 0.78 | 0.37 | 1.65 | – | – | – | – | – | – |
| | Middle | 0.51 | 0.21 | 1.24 | 0.56 | 0.22 | 1.40 | – | – | – | – | – | – |
| | Second wealthiest | **0.38** | **0.14** | **0.99** | 0.46 | 0.17 | 1.30 | – | – | – | – | – | – |
| | Wealthiest | **0.20** | **0.08** | **0.48** | **0.27** | **0.10** | **0.73** | – | – | – | – | – | – |

[1]Bolding indicates statistical significance at the p<0.05 level

safe abortion methods, including information on how to safely self-manage an abortion, are not equitably available to all groups of women. As in other legally restrictive settings, women with greater education or financial resources are able to access safer abortion care, while more vulnerable women must rely on less safe abortion methods, putting them at greater risk of abortion-related morbidity and mortality [20]; a recent study from Nigeria corroborate these findings [21]. Additionally, the methods and sources used for recent abortions suggests that the safety profile of abortions in Côte d'Ivoire is not improving; we estimate the rate of most unsafe abortions in recent years to be 25.4 based on our final incidence and safety estimates.

This study is not without limitations. The primary limitation is our inability to validate the abortion incidence or safety estimates. Although we believe the confidante measure is more accurate than the respondent estimate given concerns about underreporting with self-reported abortion data, we cannot confirm this using an external, objective measure. While other work we have done demonstrates incomplete sharing of abortions between respondents and confidantes [14], analytic decisions we made to include confidante abortions reported with less certainty helped to adjust for these biases. Another concern is that 35.9% of respondents reported no confidantes. We sought to counteract potential selection bias in the confidante data generated by the 64.1% who did report at least one confidante, however, biases may remain. Further work is needed to determine the best confidante relationship definition that optimizes representativeness of the surrogate confidante sample and respondent knowledge of confidante's abortion experiences. Our inclusion of questions on period regulation when a woman was

worried she was pregnant provided a less stigmatizing opportunity for respondents to report their own or their confidante's abortion, but it may have resulted in the inclusion of non-abortions. By averaging the pregnancy removal and combined pregnancy removal/period regulation estimates, we believe we have reduced the likelihood that our final estimate includes substantial non-abortions. However, our rates do not include repeat abortions over the 2017/2018 period, thus to the extent that Ivorian women are having repeat abortions in quick succession, our rates would be underestimates. Additionally, differential underreporting or sharing of abortions could result in inaccurate bivariate and multivariate relationships. Although not always significant, the incidence patterns across respondents and confidantes were generally similar. The failure to detect statistical significance in many relationships may have been related to limited power as a result of small sample sizes and relatively few reported abortions among some sub-groups. Related to abortion safety, differential underreporting by method and source is the primary concern. Another limitation is related to the potential for misclassification. Women were unable to provide details on the specific surgery or training of their provider, nor were many able to provide details about the specific pills they took. This could result in misclassification in both directions (e.g. facility-based surgery that was actually performed by an untrained provider, medication abortion drugs categorized as "other pills"), which would reduce the likelihood of systematic error in the end results. Lastly, although pilot results indicated correct interpretation of the pregnancy removal and period regulation phrases, the novel framing of these questions could be unintentionally capturing some miscarriages.

Our study has several strengths. Data come from a large, diverse, nationally representative sample. Our use of multiple methods to measure abortion incidence allowed us to improve the validity of incidence estimates and address potential sources of bias. In particular, use of the confidante method helped to address social desirability bias in self-reporting on abortion experiences, and adjustment for "missing" confidantes reduced likelihood of selection bias in the confidante sample. Our study provides individual, nationally representative data on abortion incidence and safety, and includes data on a range of sociodemographic characteristics, which allowed us to estimate these abortion related measures by background characteristics. A further strength of this study is our ability to document patterns of abortion incidence and safety outside the formal health care system, including self-management of abortion using pills and non-recommended methods. Considerable effort was made by the study team to develop and pilot survey questions that captured the range of women's abortion experiences and methods used in the local context, including questions on period regulation, which may not be captured in typical questions on abortion.

## Conclusion

In Côte d'Ivoire, the low prevalence of modern contraceptive use, high unmet need for family planning, and high rates of unintended pregnancy all suggest that women may frequently turn to induced abortion to manage their family size. Our findings support this, indicating that approximately 4% of women have an abortion each year, nearly two-thirds of which are unsafe. The high incidence of unsafe abortion is a significant contributor to the country's rates of maternal mortality and morbidity; Vroh et al (2012) found that 55.8% of women who reported having had an abortion also reported experiencing post-abortion complications. Consistent with other research, our results suggest that legal restrictions on abortion in Côte d'Ivoire are not keeping women from having abortions, but rather pushing women to use unsafe, potentially dangerous abortion methods. As such, efforts to address high rates of abortion and abortion-related morbidity and mortality are needed. Such measures must include ensuring access to contraceptive services offering a range of methods, quality postabortion

care services, and safe abortion for legal indications. Additionally, expanding the conditions under which women can seek safe, legal abortion has the most potential to dramatically reduce the levels of unsafe abortion and abortion-related morbidity and mortality.

## Supporting information

**S1 Doc. CIR2-female-questionnaire-English-v6-jkp.**
(PDF)

**S2 Doc. CIR2-female-questionnaire-French-v6-jkp.**
(PDF)

## Acknowledgments

We would like to acknowledge and thank our in-country team, including the Central Staff, Supervisors and interviewers who were instrumental in the data collection, as well as the women who took time to participate.

## Author Contributions

**Conceptualization:** Suzanne O. Bell, Caroline Moreau.

**Data curation:** Andoh Kouakou Hyacinthe, Georges Guiella.

**Formal analysis:** Suzanne O. Bell, Grace Sheehy.

**Funding acquisition:** Suzanne O. Bell, Caroline Moreau.

**Investigation:** Suzanne O. Bell.

**Methodology:** Suzanne O. Bell, Georges Guiella, Caroline Moreau.

**Project administration:** Suzanne O. Bell, Andoh Kouakou Hyacinthe, Georges Guiella.

**Supervision:** Andoh Kouakou Hyacinthe, Georges Guiella.

**Writing – original draft:** Suzanne O. Bell, Grace Sheehy, Caroline Moreau.

**Writing – review & editing:** Andoh Kouakou Hyacinthe, Georges Guiella.

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
