## [Decision Letter · Decision Letter 0]

21 Jan 2020

PONE-D-19-34624

The first national abortion incidence and safety estimates for Cote d'Ivoire

PLOS ONE

Dear Dr. Bell,

Thank you for submitting your manuscript to PLOS ONE. After careful consideration, we feel that it has merit but does not fully meet PLOS ONE’s publication criteria as it currently stands. Therefore, we invite you to submit a revised version of the manuscript that addresses the points raised during the review process.

ACADEMIC EDITOR: 

Your paper addresses an important theme. However, there are major content and methodological issues. I strongly encourage you carefully address the reviewers' comments.

We would appreciate receiving your revised manuscript by Mar 06 2020 11:59PM. To enhance the reproducibility of your results, we recommend that if applicable you deposit your laboratory protocols in protocols.io, where a protocol can be assigned its own identifier (DOI) such that it can be cited independently in the future. For instructions see: http://journals.plos.org/plosone/s/submission-guidelines#loc-laboratory-protocols

We look forward to receiving your revised manuscript.

Kind regards,

Luisa N. Borrell, DDS, PhD

Academic Editor

PLOS ONE

2. Please include additional information regarding the survey or questionnaire used in the study and ensure that you have provided sufficient details that others could replicate the analyses. For instance, if you developed a questionnaire as part of this study and it is not under a copyright more restrictive than CC-BY, please include a copy, in both the original language and English, as Supporting Information. Moreover, please include more details on how the questionnaire was pre-tested, and whether it was validated; and on the training of  interviewers.

3. Please provide additional details regarding participant consent. In the ethics statement in the Methods and online submission information, please ensure that you have specified (1) whether consent was informed and (2) what type you obtained (for instance, written or verbal, and if verbal, how it was documented and witnessed). If your study included minors, state whether you obtained consent from parents or guardians.

4.PLOS requires an ORCID iD for the corresponding author in Editorial Manager on papers submitted after December 6th, 2016. Please ensure that you have an ORCID iD and that it is validated in Editorial Manager. To do this, go to ‘Update my Information’ (in the upper left-hand corner of the main menu), and click on the Fetch/Validate link next to the ORCID field. This will take you to the ORCID site and allow you to create a new iD or authenticate a pre-existing iD in Editorial Manager. Please see the following video for instructions on linking an ORCID iD to your Editorial Manager account: https://www.youtube.com/watch?v=_xcclfuvtxQ

Reviewers' comments:

Reviewer's Responses to Questions

**Comments to the Author**

1. Is the manuscript technically sound, and do the data support the conclusions?

Reviewer #1: Partly

Reviewer #2: Partly

2. Has the statistical analysis been performed appropriately and rigorously? 

Reviewer #1: Yes

Reviewer #2: No

3. Have the authors made all data underlying the findings in their manuscript fully available?

Reviewer #1: Yes

Reviewer #2: Yes

4. Is the manuscript presented in an intelligible fashion and written in standard English?

Reviewer #1: Yes

Reviewer #2: Yes

5. Review Comments to the Author

Reviewer #1: Incidence of abortion and safety estimates for Cote d’Ivoire

It is extremely important to come up with estimates of abortion and abortion safety for Cote d’Ivoire. I have questions about the methodology that can likely be addressed by the authors.

Abstract

There is a missing link between abortion is common and its contributing to maternal mortality. I think you need to add that it is often unsafe. Because many countries have high abortion rates and low abortion related morbidity and mortality.

Why are these the first national estimates when you cite the Vroh study. Regardless, claims to primacy should not be in the title.

Intro

What do you mean by “unmet need”? Some estimates don’t actually ask women whether they want to be using a contraceptive. If this is the sort of measure you are using, find another. The high rate of unintended pregnancy may not be a result of unmet need for family planning but instead is a result of a low desire for childbearing.

Line 76. Is anyone ever punished? Reporting that providing abortions is punishable is not relevant if nobody is ever prosecuted for conducting an abortion. How can 47.9% of abortions take place in a health facility? Clearly what is on the law books is not the whole story.

Define unsafe on line 80.

Methods

The description of PMA 2020 as doing research on smartphones is very confusing. Did they do that for this study? Later it says face-to-face interviews.

How does the identification of a confidante work? I would like to know what question is asked – do you have a friend with whom you talk about personal matters like sex and pregnancy?

It seems very difficult to get a woman to describe the procedure that was done to her in an abortion well enough to know if the technique was sound. And certainly impossible that she should know this about a confidante’s abortion. Reports of higher unsafety among confidantes seems likely due to women just not knowing. Women don’t even necessarily know what pills they are given. How were these questions asked? “Surgery” is definitely a confusing term. I hope that wasn’t used. Real medication abortion pills (miso and mife) are safe even when provided by someone untrained.

Why was the term “pregnancy removal” used and what does it mean? Does it include miscarriage management?

Some women (apparently) don’t have confidantes. If this is real and not just an unwillingness to disclose to your interviewer, then these people are not represented in your confidante-based estimates. Do they have higher abortion rates than women with confidantes? If so, your estimate of confidante abortions is underestimated because the type of woman who has no confidantes is not represented. Or vice versa if their abortion rate is lower. Why not have the confidante’s data included along with those with no confidante. Otherwise the sort of people who don’t have friends aren’t represented. How do those with no confidantes differ in terms of willingness to disclose other sensitive info, socioeconomic circumstances, unintended pregnancy risk? It seems important to know who is omitted in the confidante estimates.

Why would the estimate of safety differ by whether it was a respondent or confidante based estimate? Is it because you learn more about the methods and training of the provider when it is the woman’s own abortion? Or is it because you only learn about your confidante’s abortion when it results in a complication?

An annual rate of 39.9 abortions per 1000 women is not the same as 4% of women getting an abortion if the rate includes multiple abortions per woman. Does it?

Typo Page 17 35.9% reported no confidantes. Not no abortions. This seems very high, by the way. What question was asked where more than a third of women have no close friends.

I think it is problematic that you do not compensate women for their time to participate. If you do, change “volunteered” their time.

Reviewer #2: January 19, 2020

Review of Manuscript: PONE-D-19-34624

The manuscript entitled “The first national abortion incidence and safety estimates for Cote d'Ivoire” aims to estimate induced abortion directly and indirectly and examines correlates of unsafe abortion, based on a population-based survey of women of reproductive age. The manuscript is largely a descriptive study and has the potential to contribute to the literature. However, it suffers from the below important methodological and structural issues.

1) Lines 104-124. The method of data collection, the structure of abortion data have not been clearly and precisely described. The authors need to provide a detailed description of method of data collection, structure of the questionnaire, sampling, and representativeness of the sample, and the survey method. I suspect that method of “smart phones” has covered all the study population; because many people are likely that they do not have a cellphone. Therefore, most likely the sample is not representative.

2) Lines 127-169: The authors’ description of “measures” of abortion rates are not at all precise, clear and useful. It is not clear how the abortion incidence rates have been calculated and what kind of data structure have been used to estimate one-year incidence rate and so on. The authors need to write in details and with precision how one-year pregnancy removal rate, one-year abortion incidence rate, combined pregnancy removal/period regulation rate were calculated.

3) The manuscript includes several levels of analysis of abortion: abortion rate estimates and correlates, and multivariate analysis of unsafe abortion. Instead of having an undeveloped manuscript with different mixed results, I would suggest the authors to break down this manuscript into two separate manuscripts: 1) estimating abortion rates; 2) studying unsafe abortion.

6. PLOS authors have the option to publish the peer review history of their article (what does this mean?). If published, this will include your full peer review and any attached files.

Reviewer #1: No

Reviewer #2: No

---

## [Author Response · Author response to Decision Letter 0]

25 Feb 2020

Dear PLOS ONE Editorial Review Committee and Reviewers,

We would like to thank you for reviewing our manuscript and for the opportunity to revise and resubmit our work for further consideration at your esteemed journal. We appreciate the thoughtful and detailed feedback that the reviewers provided, and we have done our best to incorporate the recommended changes and additions. We feel these edits have strengthened our manuscript and hope you will agree. We look forward to hearing back from you regarding your final decision.

Best,

The Authors

Journal Requirements

We updated the file naming to be consistent with journal requirements. We believe the rest of the file is formatted in accordance with PLOS ONE style requirements.

2. Please include additional information regarding the survey or questionnaire used in the study and ensure that you have provided sufficient details that others could replicate the analyses. For instance, if you developed a questionnaire as part of this study and it is not under a copyright more restrictive than CC-BY, please include a copy, in both the original language and English, as Supporting Information. Moreover, please include more details on how the questionnaire was pre-tested, and whether it was validated; and on the training of interviewers.

We have incorporated more details into the methods section, which we describe below in response to the specific reviewer comments. We also uploaded the questionnaires in English and French with the resubmission.

]3. Please provide additional details regarding participant consent. In the ethics statement in the Methods and online submission information, please ensure that you have specified (1) whether consent was informed and (2) what type you obtained (for instance, written or verbal, and if verbal, how it was documented and witnessed). If your study included minors, state whether you obtained consent from parents or guardians.

Interviewers obtained informed, verbal consent from all participants, including minors, who were treated as adults in accordance with local IRB approval. More details and the specific new text that we added to clarify this protocol are provided in response to the specific reviewer comments below.

4.PLOS requires an ORCID iD for the corresponding author in Editorial Manager on papers submitted after December 6th, 2016. Please ensure that you have an ORCID iD and that it is validated in Editorial Manager. To do this, go to ‘Update my Information’ (in the upper left-hand corner of the main menu), and click on the Fetch/Validate link next to the ORCID field. This will take you to the ORCID site and allow you to create a new iD or authenticate a pre-existing iD in Editorial Manager. Please see the following video for instructions on linking an ORCID iD to your Editorial Manager account: https://www.youtube.com/watch?v=_xcclfuvtxQ

The corresponding author has an ORCID iD (https://orcid.org/0000-0002-7650-5940) but when she tried to add it to her PLoS ONE account, it says it is already linked to an account in the PLoS ONE system. Someone at PLoS ONE will need to merge her accounts (suzannebell@jhu.edu and suzanneobell@gmail.com). 

Reviewer 1 Comments

Incidence of abortion and safety estimates for Cote d’Ivoire

It is extremely important to come up with estimates of abortion and abortion safety for Cote d’Ivoire. I have questions about the methodology that can likely be addressed by the authors.

Abstract

There is a missing link between abortion is common and its contributing to maternal mortality. I think you need to add that it is often unsafe. Because many countries have high abortion rates and low abortion related morbidity and mortality.

The reviewer makes a good point. We have modified a sentence in the background section of the abstract to clarify:“In Côte d’Ivoire, abortion is legally restricted unless a pregnancy threatens a woman’s life. Yet the limited available evidence suggests abortion is common and that unsafe abortion is contributing to the country’s high maternal mortality.”

Why are these the first national estimates when you cite the Vroh study. Regardless, claims to primacy should not be in the title.

The Vroh et al. study did not provide an abortion incidence. They only provided a lifetime prevalence and restricted the denominator to a population of women who had a child, so we could not determine the population rate from published findings. Upon the reviewer’s feedback, we changed the title to “Abortion incidence and safety in Cote d’Ivoire”.

Intro

What do you mean by “unmet need”? Some estimates don’t actually ask women whether they want to be using a contraceptive. If this is the sort of measure you are using, find another. The high rate of unintended pregnancy may not be a result of unmet need for family planning but instead is a result of a low desire for childbearing.

We were using the standard measure of unmet need from the DHS, which is an algorithm that estimates a population-level measure of unmet need using several related measures of risk of pregnancy, desire for a/another child, and use of contraception. While we acknowledge this measure is not without problems, it is a common measure in the field of demography and is useful in providing a population-level estimate that can be compared across time and geographies. However, given the reviewer’s concerns, we have removed reference to unmet need. With regard to the fact that more than one-third of pregnancies were unintended, we do not think this simply reflects a low desire for childbearing; fertility is high (total fertility rate is 5) and low desire for childbearing could be managed via contraception or avoiding sexual activity, so 35% of pregnancies being unintended is communicating something other than low desire for childbearing.

Line 76. Is anyone ever punished? Reporting that providing abortions is punishable is not relevant if nobody is ever prosecuted for conducting an abortion. How can 47.9% of abortions take place in a health facility? Clearly what is on the law books is not the whole story.

It is not uncommon for abortion to be illegal and punishable with jail time but for abortion to nonetheless be quite common. We do think it is important to describe the official law, regardless of enforcement, but we have added reference to the fact the punishment is rarely, if ever, enforced: “Despite a lack of evidence of enforcement, anyone who provides or assists in providing an abortion—as well as the woman who obtains an abortion herself—can be punished under the law with a prison sentence and fine.”

Define unsafe on line 80.

We added the WHO definition to the second half of that sentence to clarify: “West Africa has some of the highest rates of unsafe abortion in the world, with estimates indicating as many as 85% of abortions in the region are unsafe, which the World Health Organization defines as being performed by an individual lacking the necessary training or in an environment not conforming to minimal medical standards.”

Methods

The description of PMA 2020 as doing research on smartphones is very confusing. Did they do that for this study? Later it says face-to-face interviews.

PMA interviewers use smartphones to implement the survey, but the respondent is not self-administering the questionnaire. We have added the second half of this sentence to clarify: “All women age 15 to 49 were eligible to participate in the face-to-face surveys, which interviewers conducted in French or local languages using smartphones, entering data via an Open Data Kit (ODK) application on the phone.”

How does the identification of a confidante work? I would like to know what question is asked – do you have a friend with whom you talk about personal matters like sex and pregnancy?

We describe the confidante identification process and definition in the second paragraph of the Measures section within the Methods section, but we have made it more explicit that the interviewer defined the confidante relationship to the respondent: “In the abortion module, the interviewer first asked the respondent to indicate the number of close female confidantes she had, defining a close female confidante as a woman age 15 to 49 currently living in Côte d’Ivoire who shares personal information with the respondent and with whom the respondent shares personal information.”

It seems very difficult to get a woman to describe the procedure that was done to her in an abortion well enough to know if the technique was sound. And certainly impossible that she should know this about a confidante’s abortion. Reports of higher unsafety among confidantes seems likely due to women just not knowing. Women don’t even necessarily know what pills they are given. How were these questions asked? “Surgery” is definitely a confusing term. I hope that wasn’t used. Real medication abortion pills (miso and mife) are safe even when provided by someone untrained.

The reviewer makes a good point regarding not knowing the specific type and clinical context of the reported surgery, which we identify as a limitation: “Women were unable to provide details on the specific surgery or training of their provider, nor were many able to provide details about the specific pills they took. This could result in misclassification in both directions (e.g. facility-based surgery that was actually performed by an untrained provider, medication abortion drugs categorized as “other pills”), which would reduce the likelihood of systematic error in the end results.”

Regarding the “surgery” term, interviewers were not reading that option (or others) aloud; they were only selecting it if the woman described having a procedure from a provider. In the Measures section of the Methods section, we added the following additional detail: “Interviewers did not read method and source options aloud; women volunteered their answer and interviewers probed when necessary to determine the appropriate response option to select. In the pilot, women were unable to provide detail regarding the specific surgery type (e.g. dilation and curettage, manual vacuum aspiration), thus interviewers only selected the one “surgery” option if a woman described having a medical procedure.”

We agree with the reviewer that the higher level of unsafe abortions among confidantes may be related to lack of knowledge about the more safe abortions (that don’t result in complications). We explain as much in the Discussion: “Using our incidence estimate and data on the Côte d’Ivoire population, we estimate there were more than 225,000 induced abortions in 2017, the majority of which were unsafe (62.4% for respondents and 78.4% for confidantes). Since more complicated abortions are likely to be more visible to one’s social network, we view the confidante safety estimate as an overestimate. Thus, the respondent finding that 62.4% of abortions are most unsafe is more accurate.”

Why was the term “pregnancy removal” used and what does it mean? Does it include miscarriage management?

During the pilot, we tried to determine more descriptive terminology to refer to abortion instead of simply using the direct translation of “induced abortion”, which is a more stigmatizing or triggering term/phrase. We did not intend for it to include miscarriage management, although it might have. To reduce the likelihood of this, we had an initial preamble in conjunction with the first question of the abortion section that made clear we were talking about “pregnancy removal” in the context of an unintended pregnancy (“Sometimes women are worried they are pregnant or get pregnant when they do not want to be and they do something to remove the pregnancy.”). We added more detail about the piloting and intent of this phrase in the Methods section: “Terminology and question phrasing are particularly important when addressing a sensitive topic like induced abortion. In order to avoid inclusion of miscarriage experiences, interviewers read the following preamble at the outset of the abortion module to indicate that subsequent questions were in the context of an unintended pregnancy: “Sometimes women are worried they are pregnant or get pregnant when they do not want to be and they do something to remove the pregnancy.” To assess respondent interpretation, the pilot questionnaires included quantitative face validity questions where the interviewer asked the respondent to describe how she interpreted or understood the phrases “pregnancy removal” and “period regulation at a time when you were worried you were pregnant”. In total, interviewers conducted 31 pilot surveys in Cote d’Ivoire. Interviewers indicated that 100% of pilot survey respondents the “pregnancy removal” and “period regulation” phrases correctly.”

We also added reference to the possibility that we capture some miscarriages in the limitations of the Discussion section: “Lastly, although pilot results indicated correct interpretation of the pregnancy removal and period regulation phrases, the novel framing of these questions could be unintentionally capturing some miscarriages.” 

Some women (apparently) don’t have confidantes. If this is real and not just an unwillingness to disclose to your interviewer, then these people are not represented in your confidante-based estimates. Do they have higher abortion rates than women with confidantes? If so, your estimate of confidante abortions is underestimated because the type of woman who has no confidantes is not represented. Or vice versa if their abortion rate is lower. Why not have the confidante’s data included along with those with no confidante. Otherwise the sort of people who don’t have friends aren’t represented. How do those with no confidantes differ in terms of willingness to disclose other sensitive info, socioeconomic circumstances, unintended pregnancy risk? It seems important to know who is omitted in the confidante estimates.

The reviewer brings up an important potential bias, which we tried to address in the analysis (using a similar approach to what the reviewer suggested, in fact!). We have a separate confidante methods paper that provides a far more detailed description of the confidante method assumptions and adjustments taken to account for violations of method assumptions, which is currently under review. However, an early version of the paper was submitted at a conference last year and is available online (See Bell, SO. 2019. Methodological Advances in Survey-Based Abortion Measurement: Promising Findings From Nigeria, India, and Cote d’Ivoire. Population Association of America Annual Meeting, Austin, TX. Available at: http://paa2019.populationassociation.org/uploads/191027). We have provided more detail on these adjustments in the analysis section of the Methods section in the revised submission. In summary, we found that confidantes had somewhat different characteristics (likely as a result of the “missing” confidantes that correspond to the 35% of respondents who reported no confidante). We essentially imputed the likelihood of these missing confidantes having had a recent abortion using a Poisson model that regresses the characteristics of the respondents who had no confidantes, which is essentially what the reviewer is recommending; we seem to have had the same idea but hadn’t described it sufficiently! It is important to account for these “missing” confidantes as we found the respondents who had no confidantes had significantly different characteristics and were less likely to have reported their own abortion. We also constructed post-stratification weights using the respondent characteristics as a reference point in order to get the surrogate sample (i.e. the confidante data) to be representative of women of reproductive age.

Why would the estimate of safety differ by whether it was a respondent or confidante based estimate? Is it because you learn more about the methods and training of the provider when it is the woman’s own abortion? Or is it because you only learn about your confidante’s abortion when it results in a complication?

We have thought critically about the direction of the bias in comparing the respondent and confidante safety estimates. A priori, we thought the confidante estimates suffer from bias in the direction of overestimating unsafe abortions because respondents (and people generally) would be more likely to know about confidante’s abortions that resulted in complications and perhaps required the involvement and support of more people in the process of terminating the pregnancy and receiving treatment; our results support this hypothesis. Very few respondents reported a confidante’s abortion but were unable to report the method or source so that is unlikely to be the reason the confidante estimate of unsafe abortion is higher. The first paragraph of the discussion details this interpretation: “Using our incidence estimate and data on the Côte d’Ivoire population, we estimate there were more than 225,000 induced abortions in 2017, the majority of which are unsafe (62.4% for respondents and 78.4% for confidantes). Since more complicated abortions are likely to be more visible to one’s social network, we view the confidante safety estimate as an overestimate. Thus, the respondent finding that 62.4% of abortions were most unsafe is more accurate.”

An annual rate of 39.9 abortions per 1000 women is not the same as 4% of women getting an abortion if the rate includes multiple abortions per woman. Does it?

We did not ask about repeated abortions for the respondent or the confidantes. So the 4% is an accurate prevalence based on the available data. We added a sentence to the Methods section to clarify: “We did not collect information about repeated abortions thus these details correspond to the most recent pregnancy removal or period regulation.”

Typo Page 17 35.9% reported no confidantes. Not no abortions. This seems very high, by the way. What question was asked where more than a third of women have no close friends.

Thanks for catching the typo. We have addressed it. And yes, we were similarly surprised by this. We added additional clarity around the confidante definition (“a close female confidante as a woman age 15 to 49 currently living in Côte d’Ivoire who shares personal information with the respondent and with whom the respondent shares personal information”). We think the requirement around mutual sharing of personal information may have biased the relationship and caused women who do have close female friends to not report them thinking they do not meet this criterion. We raise the concern in our discussion of the limitations: “Another concern is that 35.9% of respondents reported no confidantes. We sought to counteract potential selection bias in the confidante data generated by the 64.1% who did report at least one confidante, however, biases may remain. Further work is needed to determine the best confidante relationship definition that optimizes representativeness of the surrogate confidante sample and respondent knowledge of confidante’s abortion experiences.” In future work, we want to experiment with using this phrase and a simple “closest friend” phrase to see how confidante results differ.

I think it is problematic that you do not compensate women for their time to participate. If you do, change “volunteered” their time.

Households received a small amount of phone credit (1000 CFA, which is approximately 1.6 USD), but individual women within the household did not receive an incentive for their participation. In the acknowledgement, we now say, “…as well as the women who took time to participate.”

Reviewer 2 Comments

Review of Manuscript: PONE-D-19-34624

The manuscript entitled “The first national abortion incidence and safety estimates for Cote d'Ivoire” aims to estimate induced abortion directly and indirectly and examines correlates of unsafe abortion, based on a population-based survey of women of reproductive age. The manuscript is largely a descriptive study and has the potential to contribute to the literature. However, it suffers from the below important methodological and structural issues.

1) Lines 104-124. The method of data collection, the structure of abortion data have not been clearly and precisely described. The authors need to provide a detailed description of method of data collection, structure of the questionnaire, sampling, and representativeness of the sample, and the survey method. I suspect that method of “smart phones” has covered all the study population; because many people are likely that they do not have a cellphone. Therefore, most likely the sample is not representative.

Sorry about the confusion; we obviously needed to clarify aspects of our sampling design! We used the same approach as the DHS, relying on the statistical bureau to select enumeration areas (clusters) using probability proportional to size sampling within sampling strata (i.e. urban/rural) and generating weights that represent the inverse of the probability of selection. Sampling had nothing to do with having a cell phone. Interviewers mapped and listed all households in selected clusters and invited women aged 15 to 49 in sampled households to participate. Interviewers used smartphones to conduct the face-to-face interview; respondents did not need a phone to participate, nor did they self-administer the questionnaire on a phone. This section of the methods description now reads as follows: “The sampling strategy relied on an urban-rural stratified cluster design with probability proportional to size selection of 73 enumeration areas (EAs), each of which represented a cluster of approximately 200 households. The National Statistics Institute (INS) selected the EAs from a sampling frame provided by the 2014 General Census of Population and Housing. In each EA, female resident interviewers mapped and listed all households and supervisors randomly selected 35 households from each EA sampling frame created. All women age 15 to 49 identified in selected households were eligible to participate in the face-to-face surveys, which interviewers conducted inn French or local languages using smartphones, entering data via an Open Data Kit (ODK) application on the phone; the English and the French translation of the questionnaire are provided in the supplementary materials (S1–S2). In order for the data to be nationally representative we constructed survey weights, which we calculated using the inverse of the probability of selection, accounting for the probability of EA selection, probability of household selection, and household and female response rates. The final sample included 2,738 de facto women (female response rate 98.1%). The Johns Hopkins Bloomberg School of Public Health and the Comité d'Éthique de la Recherche of Côte d’Ivoire provided ethical approval for this study. Women provided verbal informed consent prior to participation, with minors treated as adults in accordance with local IRB approval. Interviewers indicated receipt of verbal consent by checking a box in the smartphone survey to confirm receive of consent and entering their name as a witness to the consent process.” We hope this additional detail has addressed the reviewer’s concerns.

2) Lines 127-169: The authors’ description of “measures” of abortion rates are not at all precise, clear and useful. It is not clear how the abortion incidence rates have been calculated and what kind of data structure have been used to estimate one-year incidence rate and so on. The authors need to write in details and with precision how one-year pregnancy removal rate, one-year abortion incidence rate, combined pregnancy removal/period regulation rate were calculated.

We have restructured the analysis section and provided additional details on the abortion incidence calculation to improve clarity. Hopefully this addresses the reviewer’s concerns.

“We conducted univariate analyses of respondent and confidante characteristics. Due to questionnaire length constraints, we only collected information on confidantes’ age and education. We made a number of adjustments to the confidante data to improve the validity of the abortion incidence and safety estimates. The confidante estimates included all confidante pregnancy removals and period regulations that respondents reported with certainty (response option “Yes, I am certain”) or less certainty (response option “Yes, I think so”) when they could still provide the specific method(s) the confidante used. Inclusion of the less certain abortions helped to counteract respondent’s incomplete knowledge of their confidante’s abortions (i.e. transmission bias). We also adjusted the confidante estimates for potential selection bias resulting from the fact that some respondents reported zero confidantes. For this adjustment, we ran a Poisson model to predict the likelihood of these “missing” confidantes having had an abortion in the prior year. This model regressed the socioeconomic characteristics of the confidantes and the respondents with no confidantes on the available confidante abortion incidence data. We then used the model to predict the likelihood of the “missing” confidantes having had a recent abortion based on the corresponding respondents’ characteristics. We used this information to create a new variable that combined respondent reported confidante abortion data for those with confidantes, and the predicted probability of abortion in the prior year for the confidantes who were not in the sample because they had no close friends who we could have captured in the respondent sample. Research on mortality rate estimation using survey data has employed a similar modeling approach [16]. To ensure these confidante data had characteristics that reflected the population of reproductive aged women in Cote d’Ivoire, we constructed post-stratification weights using the weighted respondent data distributions as the reference. 

We calculated the one-year pregnancy removal incidence rate by determining the number of pregnancy removals reported in 2017 and in 2018 divided by the number of women in each sample. To convert the proportion into a one-year incidence rate, we divided the estimate by the total number of years covered from January 1, 2017 through the date of the interview. We then multiplied the value by 1,000 to generate the one-year estimate of pregnancy removal per 1,000 women age 15 to 49. We scaled the standard errors in the same manner. We also calculated the combined pregnancy removal and period regulation one-year incidence rate (which we refer to as “likely-abortion”). To calculate the final one-year incidence rates of induced abortion, we averaged the pregnancy removal rate and the likely-abortion rate. We averaged the two point estimates because we believe the pregnancy removal data fails to capture some abortions (that women may not view as abortions or are not willing to admit are abortions) while the period regulation data likely includes some experiences that we would not consider to be abortions. We generated all these estimates separately for respondents and confidantes. 

We calculated one-year pregnancy removal incidence rates, likely-abortion incidence rates, and the associated averages overall and by age and education for respondents and confidantes; we also calculated these rates by residence and wealth for respondents, for whom we had data on these characteristics. The final incidence analyses involved bivariate and multivariable logistic regression to determine which characteristics were independently associated with experiencing an abortion in the year prior to the survey. We used the unadjusted confidante likely-abortion dichotomous incidence data in order to conduct the logistic regression as the model assumes a Bernoulli distributed outcome variable whereas the Poisson predicted confidante incidence variable is continuous.”

3) The manuscript includes several levels of analysis of abortion: abortion rate estimates and correlates, and multivariate analysis of unsafe abortion. Instead of having an undeveloped manuscript with different mixed results, I would suggest the authors to break down this manuscript into two separate manuscripts: 1) estimating abortion rates; 2) studying unsafe abortion.

While we understand this is a lot of information, we feel it is important to include both incidence and safety estimates in this manuscript as they are related measures that together provide a comprehensive picture of abortion in Cote d’Ivoire. We hope we have laid out the content coherently, in a manner that allows the reader to easily follow the information. On a more practical level, this is a multi-country study and we do not have sufficient project funds to separate each of these country specific abortion papers into two while ensuring open access, which is a priority for us.

---

## [Decision Letter · Decision Letter 1]

25 Mar 2020

PONE-D-19-34624R1

Abortion incidence and safety in Cote d'Ivoire

PLOS ONE

Dear Dr. Bell,

Thank you for submitting your manuscript to PLOS ONE. After careful consideration, we feel that it has merit but does not fully meet PLOS ONE’s publication criteria as it currently stands. Therefore, we invite you to submit a revised version of the manuscript that addresses the points raised during the review process.

ACADEMIC EDITOR:

While the revised submission shows improvements, the reviewer has underscored major methodological concerns. Thus, I will strongly encourage you to pay careful attention to the reviewer's comments.

We would appreciate receiving your revised manuscript by May 09 2020 11:59PM. To enhance the reproducibility of your results, we recommend that if applicable you deposit your laboratory protocols in protocols.io, where a protocol can be assigned its own identifier (DOI) such that it can be cited independently in the future. For instructions see: http://journals.plos.org/plosone/s/submission-guidelines#loc-laboratory-protocols

We look forward to receiving your revised manuscript.

Kind regards,

Luisa N. Borrell, DDS, PhD

Academic Editor

PLOS ONE

Reviewers' comments:

Reviewer's Responses to Questions

**Comments to the Author**

1. If the authors have adequately addressed your comments raised in a previous round of review and you feel that this manuscript is now acceptable for publication, you may indicate that here to bypass the “Comments to the Author” section, enter your conflict of interest statement in the “Confidential to Editor” section, and submit your "Accept" recommendation.

Reviewer #2: (No Response)

2. Is the manuscript technically sound, and do the data support the conclusions?

Reviewer #2: Partly

3. Has the statistical analysis been performed appropriately and rigorously? 

Reviewer #2: (No Response)

4. Have the authors made all data underlying the findings in their manuscript fully available?

Reviewer #2: Yes

5. Is the manuscript presented in an intelligible fashion and written in standard English?

Reviewer #2: Yes

6. Review Comments to the Author

Reviewer #2: (No Response)

7. PLOS authors have the option to publish the peer review history of their article (what does this mean?). If published, this will include your full peer review and any attached files.

Reviewer #2: No

---

## [Author Response · Author response to Decision Letter 1]

6 Apr 2020

Dear PLOS ONE Editorial Review Committee and Reviewers,

We would like to thank you for a second opportunity to revise and resubmit our manuscript. We believe we have clarified and addressed the reviewer concerns and look forward to hearing back from you regarding the final decision. 

Best,

The Authors

Reviewer 2 

The revised manuscript has improved significantly; I really appreciate the authors for addressing most of the concerns that I raised in the first round of my review. However, there are serious problems in the calculation of abortion rates, based on the new description of calculation of abortion rates, which make the manuscript unpublishable if the authors cannot address them given the nature of data they have collected. 

1) Lines 232-243. Please make it clear whether the number of abortions (each specific one: pregnancy removals, period regulations, et.) in the numerator of all of the rates you calculated is “life-time” incidences or only the abortions happened in the year preceding the interview. For example, when you say, “We calculated the one-year pregnancy removal incidence rate by determining the number of pregnancy removals reported in 2017 and in 2018 divided the number of pregnancy removals reported in 2017 and in 2018”. Do you mean all pregnancy removals that a woman has had over her reproductive lifetime (say, from age 15 to the time of the interview) or the number of pregnancy removals that occurred over the year before the interview. If the number of incidences in the numerator of the calculated rate is “life-time” abortions, then your estimated rates cannot be one-year rate, you should use the number of incidences over the past year (before the date of the interview); in this case, you should have asked date of abortions in the questionnaire. Have you done this? If you clarify the structure of data that you have collected in the questionnaire can be helpful here; I don’t see English translated copy of the questionnaire in the revised manuscript. How did you determined the number of abortions in 2017 or 2018?

- From what you have written in Line *** (“The final incidence analyses involved bivariate and multivariable logistic regression to determine which characteristics were independently associated with experiencing an abortion in the year prior to the survey.”, I gather that you have calculated the abortion rates for the abortions that happened in the 12-months before the interview. If this is the case, please revise your rates based on what I have discussed above. 

Response: Our paper reports on the one-year abortion incidence rate, not lifetime incidence. The numerator for each rate is the number of events (i.e. pregnancy removal, period regulations, or the combination) in 2017 and 2018 up to the date of the interview. However, the denominator accounted for the additional 0.55 year in 2018 as we divided by the total number of woman-years so that the result, when multiplied by 1,000, represents the approximate one-year incidence rate per 1,000 women of reproductive age. We clarified this approach in the methods section: “We separately calculated the one-year pregnancy removal and period regulation incidence rates. Since we were unable to collect data on month of the event, we included events from 2017 through the date of interview in 2018. To convert this to an annualized one-year incidence rate, we divided the number of events in in 2017 and 2018 by the number of woman-years between January 1, 2017 through the date of the interview in 2018; each respondent contributed on average 1.55 woman-years. We then multiplied the value by 1,000 to generate the one-year incidence rate per 1,000 women age 15 to 49. We scaled the standard errors in the same manner.”

Since we cannot do a similar adjustment in the case of the bivariate and multivariate analyses, we reframed the bivariate/multivariate logistic regression analysis as follows to be more accurate: “The final incidence analyses involved bivariate and multivariable logistic regression to determine which characteristics were independently associated with experiencing a recent abortion in the approximately one year prior to the survey (2017 through beginning of 2018).”

The English version of the questionnaire should be attached as a supplemental file for review.

2) Line 174-175. I see here you are saying, “We did not collect information about repeated abortions thus these details correspond to the most recent pregnancy removal or period regulation.” So, did you include the most recent abortion incidence in the numerator of all incidence rates? If this is the case, therefore, you cannot claim that you have calculated “abortion rate” in this paper. Your study is limited to the pregnancy removal of the last pregnancy of women aged 15-49 in the study population. Following point 2 above, abortion rates refer to the total number of abortions that happen in a given period over the women-years of exposure to the risk of abortions in the same period. If you have not done this calculation, the section of abortion estimates is totally incorrect and unpublishable. 

Response: The reviewer brings up an important limitation that much of the survey-based abortion literature in low-resource settings also has. We know of no national estimates of repeat abortion and associated timing in Cote d’Ivoire. Data from the US suggests 48% of abortions are repeat, but the average duration between repeat abortions is 44 months (see Jones et al, 2006. Repeat abortion in the US: Occasional Report no. 29). Thus, while this would result in our incidence estimates being biased downward, we do not think the impact would be substantial. The methods section already clarifies that our data do not account for repeat abortion, but we added a sentence to the discussion section to explicitly describe this limitation: “... However, our rates do not include repeat abortions over the 2017/2018 period, thus to the extent that Ivoirian women are having repeat abortions in quick succession, our rates would be underestimates.” 

To the reviewer’s other comment regarding abortion incidence calculation, see our prior response.

3) Lines 233-235. Following my pervious concern, you said that “To convert the proportion into a one-year incidence rate, we divided the estimate by the total number of years covered from January 1, 2017 through the date of the interview.” I am wondering how you can do this when the rate you calculated is based on the abortions that happened up to 2017 or up 2018. You cannot extend it to after 2017 or 2018. To calculate an annual rate of abortion, it is correct you divide the total number of abortions that happened in a give period by 1,000 woman-years of exposure to the risk of abortion in the same given period. For example, you should divide the total number of abortions that happened in the three-year period before the interview by the women-years of exposure to the risk of abortion in the same three years.

Response: Our responses to the reviewer’s prior comments have sought to clarify how we calculated the annualized one-year incidence estimates, dividing by the total woman-years over the same period as the numerator, with each woman in the sample contributing on average 1.55 years. We hope this has addressed the reviewer’s methodological concerns.

---

## [Decision Letter · Decision Letter 2]

14 Apr 2020

Abortion incidence and safety in Cote d'Ivoire

PONE-D-19-34624R2

Dear Dr. Bell,

We are pleased to inform you that your manuscript has been judged scientifically suitable for publication and will be formally accepted for publication once it complies with all outstanding technical requirements.

With kind regards,

Luisa N. Borrell, DDS, PhD

Academic Editor

PLOS ONE

Additional Editor Comments (optional):

You have addressed the reviewers' comments satisfactorily.

Reviewers' comments:

Reviewer's Responses to Questions

**Comments to the Author**

1. If the authors have adequately addressed your comments raised in a previous round of review and you feel that this manuscript is now acceptable for publication, you may indicate that here to bypass the “Comments to the Author” section, enter your conflict of interest statement in the “Confidential to Editor” section, and submit your "Accept" recommendation.

Reviewer #2: All comments have been addressed

2. Is the manuscript technically sound, and do the data support the conclusions?

Reviewer #2: Yes

3. Has the statistical analysis been performed appropriately and rigorously? 

Reviewer #2: I Don't Know

4. Have the authors made all data underlying the findings in their manuscript fully available?

Reviewer #2: (No Response)

5. Is the manuscript presented in an intelligible fashion and written in standard English?

Reviewer #2: (No Response)

6. Review Comments to the Author

Reviewer #2: The second round of revision is satisfactory. The authors only need to acknowledge as a data limitation, that the survey did not collect complete date of abortion (only by year), so calculation of women-years exposure to abortion in the 12 months before the survey is not exact because it is only based on "year" of abortion incidence.

7. PLOS authors have the option to publish the peer review history of their article (what does this mean?). If published, this will include your full peer review and any attached files.

Reviewer #2: Yes: Professor Amir Erfani

---

## [Editor Report · Acceptance letter]

24 Apr 2020

PONE-D-19-34624R2 

Induced abortion incidence and safety in Côte d’Ivoire 

Dear Dr. Bell:

I am pleased to inform you that your manuscript has been deemed suitable for publication in PLOS ONE. Congratulations! Your manuscript is now with our production department. 

With kind regards,

on behalf of

Dr. Luisa N. Borrell 

Academic Editor

PLOS ONE